# Regenerative Effects of Heme Oxygenase Metabolites on Neuroinflammatory Diseases

**DOI:** 10.3390/ijms20010078

**Published:** 2018-12-25

**Authors:** Huiju Lee, Yoon Kyung Choi

**Affiliations:** Department of Integrative Bioscience and Biotechnology, Konkuk University, Seoul 05029, Korea; dydak01@naver.com

**Keywords:** heme oxygenase, carbon monoxide, bilirubin, neuroinflammation, regeneration

## Abstract

Heme oxygenase (HO) catabolizes heme to produce HO metabolites, such as carbon monoxide (CO) and bilirubin (BR), which have gained recognition as biological signal transduction effectors. The neurovascular unit refers to a highly evolved network among endothelial cells, pericytes, astrocytes, microglia, neurons, and neural stem cells in the central nervous system (CNS). Proper communication and functional circuitry in these diverse cell types is essential for effective CNS homeostasis. Neuroinflammation is associated with the vascular pathogenesis of many CNS disorders. CNS injury elicits responses from activated glia (e.g., astrocytes, oligodendrocytes, and microglia) and from damaged perivascular cells (e.g., pericytes and endothelial cells). Most brain lesions cause extensive proliferation and growth of existing glial cells around the site of injury, leading to reactions causing glial scarring, which may act as a major barrier to neuronal regrowth in the CNS. In addition, damaged perivascular cells lead to the breakdown of the blood-neural barrier, and an increase in immune activation, activated glia, and neuroinflammation. The present review discusses the regenerative role of HO metabolites, such as CO and BR, in various vascular diseases of the CNS such as stroke, traumatic brain injury, diabetic retinopathy, and Alzheimer’s disease, and the role of several other signaling molecules.

## 1. Overview of Heme Oxygenase Metabolites

Heme oxygenase (HO) catabolizes heme to produce biliverdin, carbon monoxide (CO), and iron (Fe^2+^). Biliverdin is a tetrapyrrolic bile pigment that is further reduced to the antioxidant bilirubin (BR) by biliverdin reductase (BVR) using NADH and NADPH as electron donors. HO plays an important role in antioxidant activity and cytoprotective action [1]. Three HO isoforms have been identified. HO-1, an inducible form of HO, is highly expressed in the spleen and other tissues involved in red blood cell degradation. HO-1 is induced in various cells by hypoxia-induced factors including stromal cell-derived factor-1 (SDF-1) and vascular endothelial growth factor (VEGF), as well as gaseous molecules such as nitric oxide (NO) and CO. The CO/HO-1 axis may further produce HO metabolites [2]. HO-2, a constitutive isoform of HO, is present in high levels in the liver, brain, and testes. In the brain, HO-2 functions as a sensor for oxygen as well as other gaseous molecules, and regulates vascular functions. HO-3 is a pseudogene found in the rat brain and does not have enzymatic activity [3,4]. NO is an endogenous gas produced by NO synthase (NOS). NO is produced by the reaction of L-arginine with three NOS isoforms: two constitutive isoforms (endothelial NOS (eNOS) and neuronal NOS (nNOS)) which react via calcium (Ca^2+^) entry, and an inducible NOS (iNOS). These enzymes are expressed in a highly cell-type-specific manner. The interplay among CO/HO-1 and NO/NOS and related signaling pathways modulates the inflammatory response and regenerative effects in many pathological situations [3,5]. Recent reports have suggested that the HO system is being explored for use in anti-inflammatory and regenerative medicine because heme metabolites suppress the release of proapoptotic and proinflammatory mediators, and facilitate mitochondrial biogenesis, resulting in tissue repair via angiogenesis and neurogenesis.

BR is converted from biliverdin by the BVR isozyme BVRA, and can easily diffuse into the lipid environment [6], acting as a signaling molecule in a paracrine manner. HO-1 expression can also be upregulated by BVRA, which is ubiquitously expressed in adult tissue, with high expression levels in the brain [7]. BVRA can act as a transcription factor and has a domain for the basic leucine zipper, the nuclear location sequence, and nuclear export sequences [8]. BVRA has the ability to bind to DNA sequences such as the antioxidant response element and hypoxia response elements [8]. On the other hand, BVRB may not have the basic leucine zipper, nuclear location sequence, and nuclear export sequences; therefore, it is unlikely that BVRB acts as a transcription factor.

HO and its metabolites are associated with cytoprotection and maintenance of homeostasis in several different organs and tissues. However, HO and its metabolites are Janus-faced. HO-1 in astrocytes of the aging and diseased central nervous system (CNS) can be an effector of deleterious stimuli, leading to neuronal injury [9]. Administration of the HO inducer hemin can increase pro-inflammatory prostaglandin E_2_ levels in rat hypothalamic explants and in primary cultures of rat hypothalamic astrocytes [10]. Conversely, the HO inhibitor Sn-protoporphyrin-9 has been proven to amplify significant activation of the hypothalamus-pituitary-adrenal axis induced by bacterial lipopolysaccharide administration in male Wistar rats [11]. This observation indicates a protective role for HO in counteracting potentially dangerous surges of serum vasopressin levels, leading to hypothalamic vasopressin depletion. Accordingly, in in vitro studies, the formation of CO within the hypothalamus has been associated with inhibition of the release of hormones, such as corticotropin-releasing hormone, arginine vasopressin, and oxytocin, involved in hypothalamus-pituitary-adrenal axis activation [12]. These findings suggest that the HO–CO pathway may have a neuroendocrine modulatory role, preventing over-activation of the hypothalamus-pituitary-adrenal axis during stress.

CO and BR may also play dual roles (i.e., pro-inflammatory and anti-inflammatory) in a variety of organs and tissues [13,14] depending on their concentration and the signaling pathways involved. Sustained activation of the HO-1-CO pathway may facilitate the development of neuroendocrine disturbances characteristic of age-related neuroinflammatory diseases [15]. To prevent neurotoxicity, BR must be glucuronidated and excreted in the bile. An excessive accumulation of BR in the brain will result in kernicteric damage. Similarly to CO and BR, free iron has a pro-oxidant capacity in a redox-active form, leading to lipid, protein, and DNA damage. Overexpression of the *HO-1* gene in oxidatively stressed astroglial cells may perpetuate intracellular reactive oxygen species (ROS) generation, oxidative mitochondrial injury, and non-transferrin-derived iron deposition within the mitochondrial compartment [16]. Free iron can have an antioxidant capacity when iron binds to ferritin, a cytosolic protein that sequesters iron in a redox-inactive form [3].

In the present review, we focus on the importance of both CO and BR in neuroinflammation associated with diseases such as stroke, traumatic brain injury (TBI), diabetic retinopathy, and Alzheimer’s disease (AD). CO and BR confer antioxidant effects by suppression of ROS and reactive nitrogen species (RNS). Concomitantly, in neuroinflammatory diseases, these HO metabolites may facilitate cell-cell communication in the neurovascular unit and bring about endogenous self-repair through regeneration such as angiogenesis and neurogenesis.

## 2. Anti-Inflammatory Role of HO Metabolites in Neuroinflammatory Diseases

Endothelial cells (ECs) communicate with pericytes, astrocytes, microglia, and neural stem cells (NSCs) in the neurovascular unit. Damage to CNS tissues breaks the blood-neural barrier by disrupting the tight junctions that link ECs [17]. This allows invasion of neutrophils and monocytes, the subsequent activation of microglia and astrocytes and, finally, the invasion of T and B cells. This inflammatory response is mediated by the nuclear factor κ-light-chain-enhancer of activated B cells (NF-κB) pathway. NF-κB controls cytokine production and cell survival, and is strongly associated with neuroinflammation in the acute phase of various vascular injuries such as stroke, TBI, diabetic retinopathy, and AD. The pro-inflammatory cytokines include tumor necrosis factor-α (TNF-α), interleukin (IL)-1, and IL-6 [18]. HO metabolites such as CO and BR play a role in anti-inflammation and cytoprotection through inhibiting production of various NF-κB-mediated cytokines under numerous neuropathological conditions [19] (Figure 1). Aside from CO gas, CO-releasing molecules (CORMs) have been developed to deliver exogenous CO to cells and tissues in a controlled manner [20]. CO can induce HO-1, leading to a HO-1/CO-positive circuit in neurovascular systems. The anti-inflammatory role of HO metabolites has been reported in various cell types in the neurovascular unit.

### 2.1. Anti-Inflammatory Role of HO Metabolites in Various Cells

#### 2.1.1. ECs

ECs induce angiogenesis to ensure a continuous supply of O_2_ and also consume O_2_ to generate signaling molecules including ROS, RNS, and NO. Low levels of NO are beneficial for cardiovascular functions such as vasodilation, inhibition of platelet aggregation, and angiogenesis [21]. Although persistently high levels of NO promote cytotoxic damage and apoptosis of ECs, physiological amounts of NO from NOS promote transcriptional HO-1 induction and CO production in vitro and in vivo [22], resulting in the promotion of vascular protection and angiogenesis. HO-1 is upregulated in ECs, and HO metabolites play a cytoprotective role against oxidative stress after vascular diseases. A BR concentration of approximately 10 μM can influence the physiology of ECs, which are in direct contact with the bloodstream. BVR-mediated BR demonstrates more potent antioxidant activity than biliverdin in ECs [23,24]. TNF-α treatment inhibits human BVR-promoter activity, which is regulated by the p65 subunit of NF-κB [25], suggesting that TNF-α may reduce BVR-mediated BR production. In human umbilical vein endothelial cells (HUVECs), CORM-2 can rescue TNF-α-mediated eNOS/NO downregulation and endothelial dysfunction by inhibiting NF-κB-responsive miR-155-5p biogenesis [19]. Exogenous BR can also partly rescue TNF-α-mediated eNOS downregulation [19], implying the antioxidant effects of BVR on the vascular system. HO-1 or BVR knock-down results in increased cellular oxidative damage in vascular diseases based on lipopolysaccharide-triggered RNS formation in HUVECs [24]. The antioxidant effects of CO in synergistic action with BR are partially based on direct ROS scavenging but also on downregulation of NADPH oxidase activity, neutrophil adhesion, and NF-κB pathways [23]. Taken together, the HO-1/CO pathway in ECs leads to decreased proinflammatory responses partly via BVR-mediated BR production.

#### 2.1.2. Pericytes

Pericytes are located on the capillary wall and share a common basement membrane with ECs in the CNS and the peripheral nervous system. Interestingly, CNS ECs have higher pericyte coverage than peripheral tissues [26]. The association between CNS endothelium and pericytes is important for maintenance of adherence and tight junctions, blood-neural barriers, and homeostasis [17,27,28]. In the retina, HO-2 has been detected in the inner segment of photoreceptors, in amacrine cells, some bipolar and ganglion cells, and in some astrocytes [29], implying that CO and BR are synthesized by HO-2 and can diffuse into neighboring cells in physiological conditions in the retina. In pathologic conditions, CO induces the nuclear translocation and accumulation of nuclear factor erythroid 2-related factor 2 (Nrf2) [30] and HO-1 is upregulated via Nrf2 after retinal ischemic injury [31].

Retinal pericytes―not retinal ECs―undergo accelerated NF-κB-mediated apoptosis in the presence of high glucose levels (a diabetic environment) [32]. Treatment of bovine retinal pericyte cells with insulin abrogates the effect of H_2_O_2_ on pericyte apoptosis [33]. Activated NF-κB in retinal pericytes upregulates TNF-α expression, which then generates apoptotic and inflammatory signals [32]. Pericyte loss can be also observed in mouse models of TBI [5]. Pericyte degeneration can lead to white matter disruption, resulting in a loss of myelin, axons, and oligodendrocytes, consequently reducing working memory [28,34]. CORM-3 injection has been seen to markedly reduce pericyte apoptosis, possibly by inhibition of hypoxia-inducible factor-1α (HIF-1α) in TBI brain pericytes [5] (Figure 1). HO metabolites may therefore play a beneficial role in pathophysiological conditions in the retina by rescuing pericyte cell death and cellular communications.

#### 2.1.3. Astrocytes

Astrocytes are classified into one of the major glia cell types and play essential roles in the regulation of neurodevelopment, neurometabolism, elimination of excess synapses through phagocytosis, and angiogenesis [35,36,37,38]. Neuroinflammation and ischemia induce different types of reactive astrocytes (e.g., A1 and A2) [35]. A1 neuroinflammatory reactive astrocytes may be induced by NF-κB signaling and have harmful functions which lead to the destruction of synapses [39]. A2 reactive astrocytes can be induced by ischemia and promote glial scarring (Figure 1). Glial scarring has beneficial effects that encapsulate the site of inflammation to form a protective barrier for adjacent healthy brain tissue. The proliferation and hypertrophy of glial cells, however, leads to increased expression of chemorepellents (e.g., semaphorin 3A) or growth-inhibiting molecules that inhibit axon growth and tube formation in ECs [40]. Hence, it could be beneficial to develop medicines that minimize neuroinflammation, form a proper protective barrier, and provide an environment that supports the proliferation of multipotent NSCs and ECs. CORM-3-injected mice brains exhibit less GFAP expression compared with TBI control brains, suggesting that CO may reduce hypertrophy of astrocytes [5].

Angiogenic and energy-sensing genes in astrocytes can be produced by HO metabolites [41]. HO-1-mediated CO and BR induction stabilizes peroxisome proliferators-activated receptor γ-coactivator-1α (PGC-1α) and HIF-1α in astrocytes, which play a key role in mitochondria biogenesis, angiogenesis via estrogen-related receptor α (ERRα), and VEGF in ischemic brain injury [41,42,43]. Moreover, astrocytes can produce functional extracellular mitochondria and transfer mitochondria to injured neurons after ischemic stroke [38]. In this way, HO metabolites may influence astrocytic functions by the reduction of inflammatory reactivation via the enhancement of mitochondrial biogenesis.

#### 2.1.4. Neurons

The energy requirements of the brain and retina are very high, and glucose is the obligatory energy substrate of the adult brain and retina. Because neurons account for most of the energy consumption during brain and retina activation, neuronal activity-dependent (somatosensory and visual) increases in blood flow and glucose utilization are matched by increases in O_2_ consumption [44]. Neurometabolism is an oxidative process characterized by a series of metabolic steps resulting in O_2_-dependent ATP production in the mitochondria. In the presence of oxidative stress in neurons, HO metabolites may suppress inflammatory responses. Dore et al. have demonstrated that accumulation of BR due to enhancement of HO-2 catalytic activity is neuroprotective against H_2_O_2_-induced cytotoxicity in neuronal cultures [45,46]. The neuroprotective effect conferred by HO-1 has been demonstrated when neuron-specific HO-1 overexpression was observed to result in reduced glutamate toxicity as well as a reduction in H_2_O_2_-induced cell death [47]. HO-1 activity is regulated by Nrf2 or/and BVR in neuronal cells [48,49]. Therefore, the Nrf2/HO-1 and BVR/HO-1 pathway may contribute to an essential defense mechanism for neurons exposed to oxidant stress via HO metabolites such as CO and BR.

### 2.2. Anti-Inflammatory Role of HO Metabolites in Neuroinflammatory Diseases

#### 2.2.1. Stroke

Glia-neuron interactions play key roles in the regulation of cerebral blood flow, energy supply, and repair processes [38,44]. Stroke can lead to neuronal death and reactive glia cells, disrupting glia-neuron interactions. In reactive glia, increases in neurotoxic molecules such as IL-1β and TNF-α induce HO-1 mRNA expression in conditions of oxidative stress which in turn leads to mitochondrial damage [9,50]. In transgenic mice overexpressing human HO-1 in their astrocytes, HO metabolites promote mitochondrial sequestration of non-transferrin iron, oxidative stress-mediated substrate modification within the mitochondria, and subsequent mitophagy [50]. Co-culture of PC12 cells with HO-1 overexpressing astrocytes induces PC12 cell death, which is reduced by treatment with deferoxamine, an iron chelating agent [51]. Therefore, the neurotoxic role of HO-1 in oxidative stress-conditioned astrocytes may stem from excessive iron deposition.

CORM-3 promotes neuroprotection by reducing the levels of inflammatory factors such as TNFα when CORM-3 is administered either before or three days after intracerebral hemorrhage. More specifically, it will reduce these levels as a prophylactic agent or during the post-acute inflammatory phase [52]. In addition, CORM-3-induced HO-1 upregulation protects rat astrocyte cells from IL-1β-mediated inflammatory responses and cell migration [53]. HO-1 transgenic mice demonstrate smaller infarct volumes, increased levels of anti-apoptotic proteins, and diminished tissue lipid peroxidation compared with wildtype mice experiencing cerebral ischemia [54]. HO-1 is an upstream factor for HIF-1α stabilization in the periinfarct region [42] and HIF-1α induction in ischemic neurons may play a therapeutic role by inducing VEGF in a stroke model [55,56]. HIF-1α can induce BVR expression [57] and BVR levels are increased in neuronal populations with HO-1 expression in ischemic injury [58]. Therefore, BVR-HO-1-HIF-1α circuitry in neurons may play a role in neuroprotection in ischemic stroke.

Beyond the prevention of neuronal death in stroke, HO metabolites (i.e., CO and BR) may be capable of facilitating repair and regeneration. Transfer of astrocytic-derived mitochondria into neurons may enhance neuroplasticity for behavioral improvement [38]. In recent studies, astrocytic HO-1 induction has been observed to increase HIF-1α, PGC-1α, and ERRα levels in a mouse model of ischemia/reperfusion injury [41,42,59]. The HO-1 metabolites CO and BR have been seen to enhance HIF-1α stabilization in astrocytes by eliciting sequential activation of the CaMKKβ/AMPKα/PGC-1α/ERRα axis. This pathway results in increases in mitochondrial biogenesis and oxygen consumption, and consequent transient intracellular hypoxia that stabilizes the HIF-1α protein. Activation of HIF-1α in astrocytes can lead to transfer of mitochondria into neurons, and angiogenesis after ischemic injury. In this way, enhanced glia-neuron and glia-vasculature interactions mediated by HO metabolites can contribute to delayed neuroplasticity in stroke outcomes.

#### 2.2.2. TBI

TBI can lead to degenerative brain damage, which results in mood, memory, and cognitive difficulties. After TBI, a glutamate wave excites neurons via excess Ca^2+^ entry. Ca^2+^-mediated NOS activation and consequent peroxynitrite (ONOO^−^) production may lead to cytotoxicity in CNS cells [60]. The link between BR and NO has been reported as a determinant of neuroinflammation [61]. BR upregulates the nNOS/NO axis in primary rat cerebellar granule neurons exposed to serum starvation or conditions of neurotrophin deficiency [61]. In the absence of exogenous stimuli, BR upregulates the phosphorylation of the cAMP responsive element binding (CREB) factor and the production of NO. The extracellular Ca^2+^ chelator ethylene glycol-*bis*(2-aminoethylether)-*N,N,N′,N′*-tetraacetic acid (EGTA) interferes with this pathway [62], suggesting the role of BR in Ca^2+^-mediated CREB and nNOS activation. Both CREB and NO are considered important factors contributing to synaptic plasticity and memory consolidation, as they regulate the expression of brain-derived neurotrophic factor (BDNF) [63,64]. BR may therefore boost the repair process, counteracting the deficiency of neurotrophic factors following brain injury. On the other hand, treatment of PC12 cells with BDNF or neurotrophic growth factor increases signaling to Akt (Protein Kinase B) and extracellular signal-regulated kinases (ERKs), which are crucial factors for survival, and these effects are markedly reduced by BR [62]. These observations indicate an important effect of BR on survival signaling mediated by neurotrophins, with either inhibitory or agonistic effects based on growth factor availability.

In addition, BR serves as an endogenous scavenger for RNS by denitrosylating the thiol group of proteins and non-protein molecules [65]. In response to exogenous hydrogen peroxide, BR markedly decreases ROS generation in PC12 cells [62]. BR can be formed by the HO-BVR pathway, and HO is co-expressed with BVR in the brain [66]. Therefore, BR may reduce peroxynitrite production when the HO-BVR pathway is activated in the TBI brain. One possible target that may lead to activation of the HO-1-BVR pathway in TBI is CO.

In one study, an apoptotic marker, cleaved caspase 3, was co-stained with pericyte marker PDGFR-β in ipsilateral mice brains on day three after TBI, and was significantly reduced by CORM-3 or CO gas [5]. In addition, CO was seen to suppress ROS production in pericytes, consequently protecting neighboring cells from RNS-mediated cytotoxicity in a paracrine manner. CORM-3-treated TBI brains exhibited less expression of GFAP and higher expression of NeuN (a mature neuronal marker) compared with TBI mice brains at day 21 after TBI, and these effects were significantly reversed by NOS inhibition [5], suggesting that CO-NO crosstalk may reduce glial scarring and facilitate neuronal regeneration (Figure 1).

In addition, CO may enhance crosstalk between pericyte-NSCs in the delayed neuroplasticity phase of TBI. NSCs located in CORM-3-treated TBI mice brains have exhibited increases in their ability to synthesize, migrate, and differentiate into mature neurons compared with those in TBI mice brains [5]. NSCs are in close proximity to pericytes in the CORM-3-treated TBI brain, and the enhanced in vitro differentiation of NSCs into mature neurons has been observed when NSCs are incubated with a conditioned medium from CORM-3-treated pericytes under oxygen-glucose deprivation [5]. CO may therefore provide a therapeutic approach to TBI by reducing pericyte death, facilitating cross-talk with NSCs, and promoting NOS/NO-mediated neurogenesis.

#### 2.2.3. Diabetic Retinopathy

In patients with diabetes, hyperglycemia-driven excess generation of ROS induces oxidative stress in a variety of tissues [3]. Oxidative stress is closely associated with chronic inflammation and plays a key role in the pathogenesis of vascular complications. Diabetic retinopathy is a major complication of diabetes and is a leading cause of blindness. Diabetic retinopathy is characterized by increased blood-retinal barrier breakdown, edema, and EC and pericyte cell death [67]. 

HO-2 and nNOS immunoreactivity has been detected in many retinal neurons, which strongly suggests that interactions between CO and NO may occur [68,69]. CO can amplify the NO-induced levels of cyclic guanosine monophosphate (cGMP) in large numbers and types of retinal neurons. Application of NOS inhibitors has blocked increases in cGMP induced by endogenous CO [70]. These results support CO as a possible modulator of the guanylyl cyclase/cGMP signaling pathway in photoreceptors of the retina (Figure 2A). 

In addition, neuronal/glial degenerative changes have been demonstrated in diabetic retinopathy [71]. Müller cells are one of the principle glia of the retina and are a source of retinal stem cells. Müller cells can be reprogrammed to generate rod photoreceptors, leading to restored visual responses in a mouse model of congenital blindness [72]. VEGF receptor 2 (VEGFR2) knockout mice have exhibited a gradual reduction in Müller cell density in the diabetic retina compared with that in wildtype mice [73]. VEGFR2-mediated Müller cell survival is required for the viability of retinal neurons in diabetic retina [73]. HO-1 can be detected in rat Müller cells and promotes the survival of Müller cells after ischemia-reperfusion injury, which is abolished by HO-1 siRNA injection intravitreally before ischemia [69]. HO-1 siRNA-treated retina on day 14 after reperfusion has demonstrated increased macrophage infiltration and severe destruction of the retinal architecture [69]. As a result, the regenerative effects of Müller cells may be regulated by HO metabolites in diabetic retinopathy.

Serum BR level is a strong protective factor against, and a possible candidate biomarker for, many complications of vascular diseases such as diabetes and atherosclerosis, or multiple sclerosis [74,75,76]. Additionally, total BR levels in plasma have been negatively related to the incidence of arterial stiffness in type 2 diabetes mellitus [77]. In experimental studies, it has been demonstrated that the beneficial effect of HO-1 induction results from an increase in PPARα. It has also been reported that that BR binds directly to the nuclear receptor transcription factor PPARα to reduce adiposity and blood glucose levels [78]. Therefore, the HO metabolites CO and BR may rescue neuronal/glial degenerative changes by reducing oxidative stress and promoting cellular networking among various cells in the retina (Figure 2b).

#### 2.2.4. AD

In patients with AD, failure of recent memory and other intellectual functions has been observed. Amyloid precursor protein (APP) generates the β-amyloid (Aβ) peptide, which has been postulated to participate in the neurotoxicity found in AD. Neuronal loss and reactive astrocytes may be associated with Aβ-peptide toxicity and the deposition of neurofibrillary tangles containing hyperphosphorylated tau in AD [9]. HO-2 interacts with APP and APP inhibits HO activity [79]. Treatment of rat hippocampal neurons with CORM-2 has been observed to protect them against Aβ-induced toxicity [80]. A marked reduction in neuronal BR levels may be related to an increased sensitivity to H_2_O_2_-induced neurotoxicity in transgenic APP_swe_ mice [79], suggesting that HO metabolites such as CO and BR may play cytoprotective roles in AD. Similar to TBI, pericyte loss is also promoted in AD, consequently leading to the accumulation of neurotoxic Aβ in transgenic APP_swe_ mice [81]. The link between HO metabolites and functional pericyte recovery is of great interest with regard to AD.

Recently, a down-regulation of HO-2 and BVR has been demonstrated in post-mortem brain tissues of AD subjects as compared to tissues of age-matched controls. These changes were found in the hippocampus, an area associated with cognitive functions such as learning and memory [82,83,84]. Moreover, a significant increase in the phosphorylation of HO-1 serine residues was observed in the hippocampus of AD subjects [84]. As a result of phosphorylation, HO-1 can become a target for oxidative posttranslational modifications of the protein structure, with consequent functional impairment. In addition, repression of HO-1 mRNA has been reported under strong and sustained pro-oxidant conditions such as hypoxia, heat shock, and interferon-γ [85,86,87]. Treatment of HUVECs with the iron chelator desferrioxamine, or hypoxia, has been seen to reduce HO-1 mRNA expression [87], possibly preventing toxic accumulation of HO metabolites such as iron and CO. Though HO-1 mRNA expression in plasma has been reported to be markedly reduced in AD subjects in comparison with healthy controls [88], recent studies show enhanced HO-1 protein levels in the plasma, hippocampus, and cerebellum of AD subjects [84,89]. For this reason, reports of HO-1 levels in AD are still controversial.

3xTg-AD mice have three mutant human genes—APP_swe_**,** PS1_M146V_, and tau_P301L_—and show reductions in BVRA protein levels at three, six, and 12 months old [83]. BVRA can be modified by oxidative or nitrosative stress into tyrosine nitration (3-NT-BVRA) which is increased in the hippocampus of 12 and 18 month-old 3xTg-AD mice compared to wildtype mice [83]. Impaired BVRA activity, partly via 3-NT-BVRA, and reduction of BVRA expression may result in increased oxidative stress and inflammation in AD [83,90].

Administration of the cholesterol-lowering agent atorvastatin can induce BVRA and HO-1 protein expression in the parietal cortex of aging dogs [91,92,93], resulting in increased BR levels. In addition, atorvastatin has been observed to lower oxidative/nitrosative stress biomarkers, such as 3-nitrotyrosine, in the same cortical area [92,93], leading to the reduction of oxidative stress-mediated neuroinflammation. Size discrimination learning error scores have negatively correlated with BVRA protein levels and BVR activity [92]. Therefore, atorvastatin-mediated HO-1 and BVR upregulation may be associated with reduced oxidative stress and improved cognitive functions in the aging brain.

## 3. Regenerative Effects of HO Metabolites

Neurons in the adult human brain have only a limited capacity for regeneration. Nevertheless, neuron replacement by endogenous NSCs may be able to repair the injured brain under favorable conditions. In humans, the entire subventricular zone of the cerebral hemispheres near the vessels and the hippocampal dentate gyrus provide a supportive environment for NSCs. A cellular network comprising NSCs, ECs, pericytes, microglia, and astrocytes may enhance the environment for angiogenesis and neurogenesis. The distinct role of HO metabolites in neurovascular functions, such as angiogenesis and neurogenesis, has been demonstrated. The ensuing sections discuss the important role of HO metabolites in angiogenic and neurogenic functions, and factors such as VEGF, BDNF, SDF-1, and osteopontin (OPN) during neurovascular remodeling and repair by the stimulation of cell–cell communications. These signals can activate phosphatidylinositide 3-kinases (PI3K)-Akt and mitogen-activated protein kinase (MEK)-ERK1/2 pathways through various transmembrane receptors [94,95,96], consequently inducing angiogenesis and neurogenesis (Figure 3).

### 3.1. VEGF

Genetic overexpression of *HO-1* enhances VEGF synthesis and secretion in astrocytes, leading to angiogenic effects on HUVECs and brain ECs by binding to its receptor, VEGFR [41,43]. VEGF augments the formation of vascular capillaries, improving blood flow in ischemic tissues; this effect is blocked by treating animal subjects with an HO inhibitor, indicating that HO-1-induced VEGF expression and angiogenesis are specifically influenced by HO metabolites [97,98]. In one study, two groups of mice―vehicle-treated and BR-treated―were subjected to unilateral hindlimb ischemia surgery. The BR-treated group exhibited enhancement of blood flow recovery in response to ischemia through the eNOS and Akt pathways [99]. The BR-mediated Akt pathway may also play a role in regeneration via cross-talk with the eNOS/NO pathway in ischemic injury.

HIF-1α, PGC-1α, and ERRα are associated with expression of VEGF and mitochondrial genes, consequently inducing angiogenesis and mitochondria biogenesis [37,43,100]. Astrocytes play a key role in maintaining vascular function and neuroprotection in ischemic diseases [38,101] by increasing neurogenic and angiogenic factors including VEGF [102,103]. The HO-1/CO pathway fulfills various important roles in angiogenesis and mitochondria biogenesis in astrocytes following cerebral ischemia in mice [41,59]. Combined CO and BR has been observed to increase HIF-1α stability by sequential activation of the L-type Ca^2^^+^ channel, Ca^2^^+^/calmodulin-dependent protein kinase β, and AMP-activated protein kinase α, resulting in increases in mitochondrial O_2_ consumption. Stabilization of HIF-1α further increases VEGF and ERRα expression at the transcriptional level, implying a positive circuit between HIF-1α and ERRα, consequently increasing VEGF [42]. Thus, HO metabolites may contribute to both angiogenesis and energy metabolism, leading to potential improvement of neurovascular function after ischemic or hypoxic injury.

### 3.2. BDNF

The HO metabolites CO and BR modulate BDNF levels in glia and neurons [104]. BDNF promotes neuronal survival, differentiation, synaptic plasticity, and angiogenesis in pathophysiological brains [105]. In one study, the human HO-1 gene was locally injected into rat substantia nigra concomitantly with 1-methyl-4-phenylpyridinium, a dopaminergic neurotoxin. Seven days after injection, the 1-methyl-4-phenylpyridinium and *HO-1* overexpression was observed to have increased the survival rate of dopaminergic neurons, reduced the production of TNF-α and IL-1β in substantia nigra, antagonized the reduction of striatal dopamine content induced by 1-methyl-4-phenylpyridinium, and up-regulated BDNF expression in the substantia nigra [106]. BDNF significantly increased expression levels of the tropomyosin-related kinase B (TrkB) receptor gene and phosphorylation of the TrkB protein in the lesions. BDNF and its high-affinity receptor TrkB were expressed in the developing mouse cerebral cortex [107]. Overexpression of *HO-1* using plasmid injection through hippocampal CA1 injection five days before a cerebral ischemia/reperfusion rat model was employed was seen to result in the activation of the BDNF-TrkB-PI3K/Akt signaling pathway [108]. NSCs genetically modified to encode the BDNF gene further enhanced synaptogenesis; BDNF in NSCs or naive NSCs were directly engrafted into lesions in a rat model of TBI. The expression levels of Ras, phosphorylated ERK1/2, and postsynaptic density protein-95 were elevated in the BDNF/NSCs-transplanted groups compared with those in the NSCs-transplanted groups throughout the experimental period. Moreover, the Nrf2/thioredoxin axis, which is a specific therapeutic target for the treatment of injury or cell death, was upregulated by BDNF overexpression [109]. Therefore, in mature neural circuits, HO-1-BDNF-TrkB signaling may modulate synaptic efficacy and synaptic plasticity, including long-term potentiation and long-term depression. 

### 3.3. Stromal Cell-Derived Factor-1 (SDF-1)

Aortic rings isolated from HO-1-deficient mice are unable to form capillary sprouts ex vivo in response to the chemokine SDF-1, which plays a role in the recruitment of endothelial progenitor cells to sites of injury and to facilitate repair. This defect has been seen to be reversed by treatment with CORM-2 but not with BR [110], indicating that vascular remodeling by SDF-1 requires HO-1 induction and subsequent CO production. Therefore, HO-1/CO must be closely linked to the proangiogenic effects of SDF-1. Recruitment of bone marrow-derived stem cells into the degenerating retina has been found in intravitreal injection of SDF-1, which was associated with decreased activated microglia and improved visual function [111]. NSCs have been observed to be recruited to sites of CNS injury by the cytokine SDF-1α that is expressed in the damaged tissues, acting on the CXC chemokine receptor type 4 (CXCR4). BDNF pretreatment of NSCs results in significantly greater migration to SDF-1 and expression of CXCR4 [112]. Hence, CO-induced SDF-1 may have the ability to elicit CXCR4-expressing NSCs to migrate into injured tissues, partly through BDNF, leading to regenerative effects.

### 3.4. NOS/NO

CO and NO, produced by catalytic reactions from HO and NOS, share similar properties. They are highly diffusible, and these gaseous molecules affect neighboring cells without acting through a transmembrane receptor. CO and NO are extremely short-lived, and can therefore serve as transcellular messengers by modulating second messengers such as cGMP and by binding various proteins [2].

Like CO, NO also induces angiogenesis partly through HO-1-mediated VEGF induction. Treatment of ECs with an NO donor enhances the protein levels of VEGF and HO-1 [2]. HIF-1α induction in response to ischemia causes the release of VEGF, leading to an increase in NO production, and EC migration and proliferation to promote angiogenesis [37,113]. In addition to angiogenesis, nNOS/NO can act as an intracellular signal that regulates neurogenesis in mouse brain neural progenitor cells. BDNF increases nNOS protein levels, and nNOS/NO has the ability to induce the maturation of neurons from neural progenitor cells [107]. Injection of CORM-3 in TBI mice has produced enhanced proliferation and migration of adult NSCs in concert with behavioral improvement, which can be markedly reduced by NOS inhibitors in vivo [5]. The effect of the HO metabolite pathway on the neurovascular system can cross-talk with the NOS/NO pathway, which may regulate the expression of angiogenic and neurogenic factors.

### 3.5. OPN

OPN is a 314 amino acid glycoprotein, and its secreted form can be involved in cell attachment and signaling. OPN can bind to extracellular matrix proteins, fibronectin, and collagen [114]. An increase in secreted OPN levels has also been found in CO-treated cultures. Short-term exposure to 25 parts per million CO at days 1 and 4 has been seen to significantly increase the relative content of β-tubulin III-immunoreactive immature neurons and tyrosine-hydroxylase-expressing catecholaminergic neurons, as assessed 5–6 days after differentiation from human NSCs [115]. The HO-1-OPN axis mediates cell proliferation and odontoblast differentiation in human dental pulp cells [116].

OPN is also an angiogenic factor, inducing EC proliferation and migration. Hypoxia-driven OPN stimulates HIF-1α-mediated VEGF expression via PI3K/Akt and ERK pathways. In addition, VEGF induces OPN with the OPN receptor integrin αvβ3 [117,118]. The anti-OPN or anti-αvβ3 antibody has more anti-angiogenic effects compared with the anti-VEGF antibody in vivo [118]. Similar to SDF-1α, OPN binds to CXCR4 and promotes NSCs migration [119]. Binding of OPN with its receptor, CD44, is crucial for many cellular functions such as angiogenesis [120]. As a result, secreted OPN in hypoxic conditions binds to several receptors and forms a positive circuit with VEGF, consequently promoting angiogenesis and NSC migration.

## 4. Perspectives

Neurovascular coordination refers to the highly evolved network comprising ECs, pericytes, astrocytes, microglia, neurons, and NSCs in the CNS. Proper communication and functional circuitry of these diverse cell types is essential for effective CNS homeostasis. In neuroinflammatory diseases, the cellular network is disrupted. HO metabolites, such as CO and BR, have the ability to reduce inflammatory responses by inhibiting NF-κB-mediated cytokine production and diminishing the generation of ROS and RNS. Cells that survive on account of the effects of HO metabolites can contribute to endogenous repair of the cellular network by releasing neurovascular regeneration factors including VEGF, BDNF, SDF-1, NO, and OPN. These plentiful cellular networks result from diverse neurotrophic factors secreted from neurovascular cells that have the powerful capacity to regenerate functional vessels and neurons in CNS diseases such as stroke, TBI, diabetic retinopathy, and AD, subsequently leading to self-repair of the functional neural circuitry (Figure 4). However, sustained upregulation of HO and its metabolites in the brain may exacerbate neural injury. The disparate properties of HO and its metabolites under various neuropathological conditions must therefore be carefully considered for clinical approaches. 

## Figures and Tables

**Figure 1 ijms-20-00078-f001:**
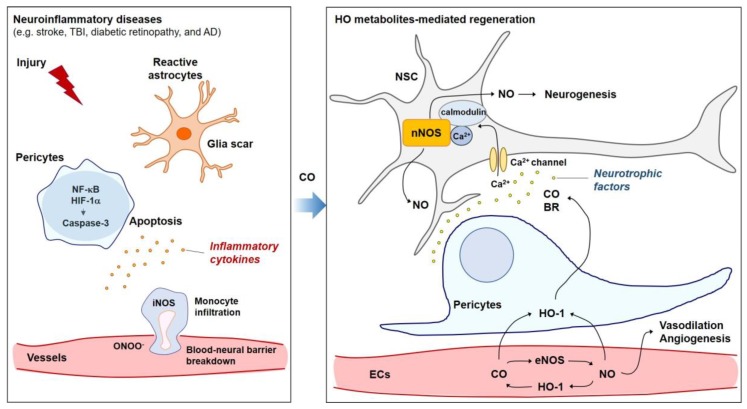
Inflammatory response is mediated by the nuclear factor κ-light-chain-enhancer of activated B cells (NF-κB), hypoxia-inducible factor (HIF)-1α, and reactive nitrogen species (RNS (e.g., ONOO^−^)) production. This is strongly associated with glia scar formation, pericyte cell death, and invasion of monocytes in the acute phase of various vascular injuries such as stroke, traumatic brain injury (TBI), diabetic retinopathy, and Alzheimer’s disease (AD). Heme oxygenase (HO) metabolites, such as carbon monoxide (CO) and bilirubin (BR) play a role in anti-inflammation and cytoprotection through inhibiting the NF-κB-mediated production of various cytokines. After CO treatment, CO-mediated HO-1 induction in endothelial cells (ECs) may facilitate angiogenesis partly via crosstalk between HO/CO and endothelial nitric oxide synthase (eNOS)/nitric oxide (NO). In addition, the HO metabolites CO and BR may stimulate neurogenesis partly via a Ca^2+^/calmodulin-mediated neuronal NOS (nNOS)/NO pathway in neural stem cells (NSCs).

**Figure 2 ijms-20-00078-f002:**
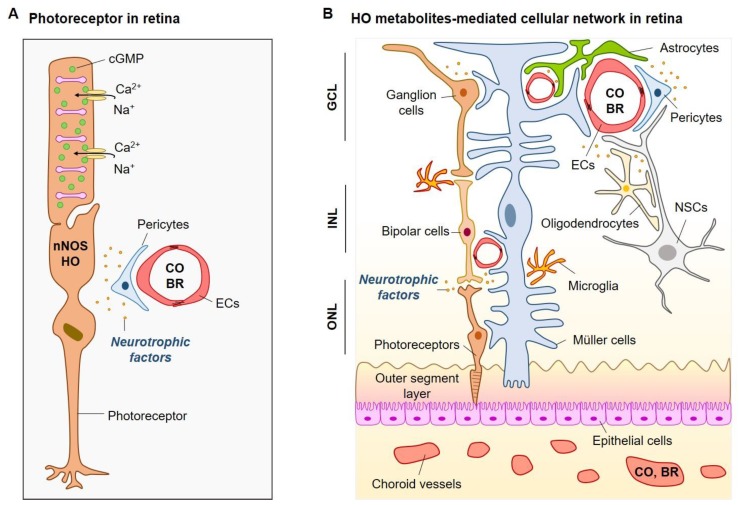
Possible role of HO metabolites in the retina: (**A**) HO metabolites may activate cyclic guanosine monophosphate (cGMP)-gated channels in photoreceptors partly via crosstalk between perivascular cells and photoreceptors. In the presence of HO metabolites, cGMP levels in the outer segment membrane may be high; cGMP binds to cation (Na^+^ and Ca^2+^)-permeable channels in the membrane, keeping them open and depolarizing the photoreceptor. (**B**) The complex architecture of the retina may be strengthened by HO metabolites. A neuronal chain from the photoreceptors to the bipolar cells to the ganglion cells provides the most direct route for transmitting visual information to the brain. Cellular networks comprising various retinal cells (e.g., Müller cells, ECs, pericytes, NSCs, oligodendrocytes, and microglia) can lead to self-repair by regenerating retinal neurons and vessels after retinal injury by HO metabolite-mediated neurotrophic factors. ONL is the outer nuclear layer, INL is the inner nuclear layer, and GCL is the ganglion cell layer.

**Figure 3 ijms-20-00078-f003:**
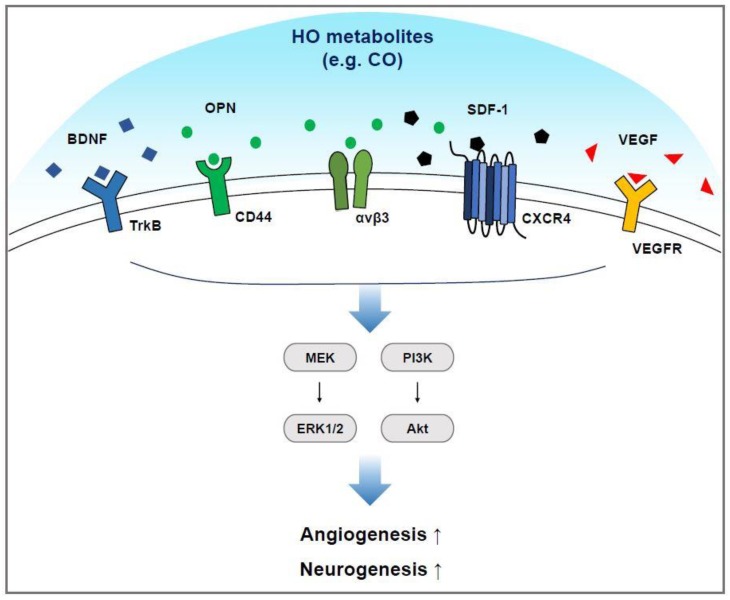
HO metabolites such as CO, can lead to the secretion of various factors (e.g. BDNF, OPN, SDF-1, and VEGF) associated with the regenerative machinery. These factors can bind to their specific receptors, stimulating signaling axes such as MEK-ERK1/2, and PI3K-Akt in ECs and NSCs, possibly resulting in angiogenesis and neurogenesis.

**Figure 4 ijms-20-00078-f004:**
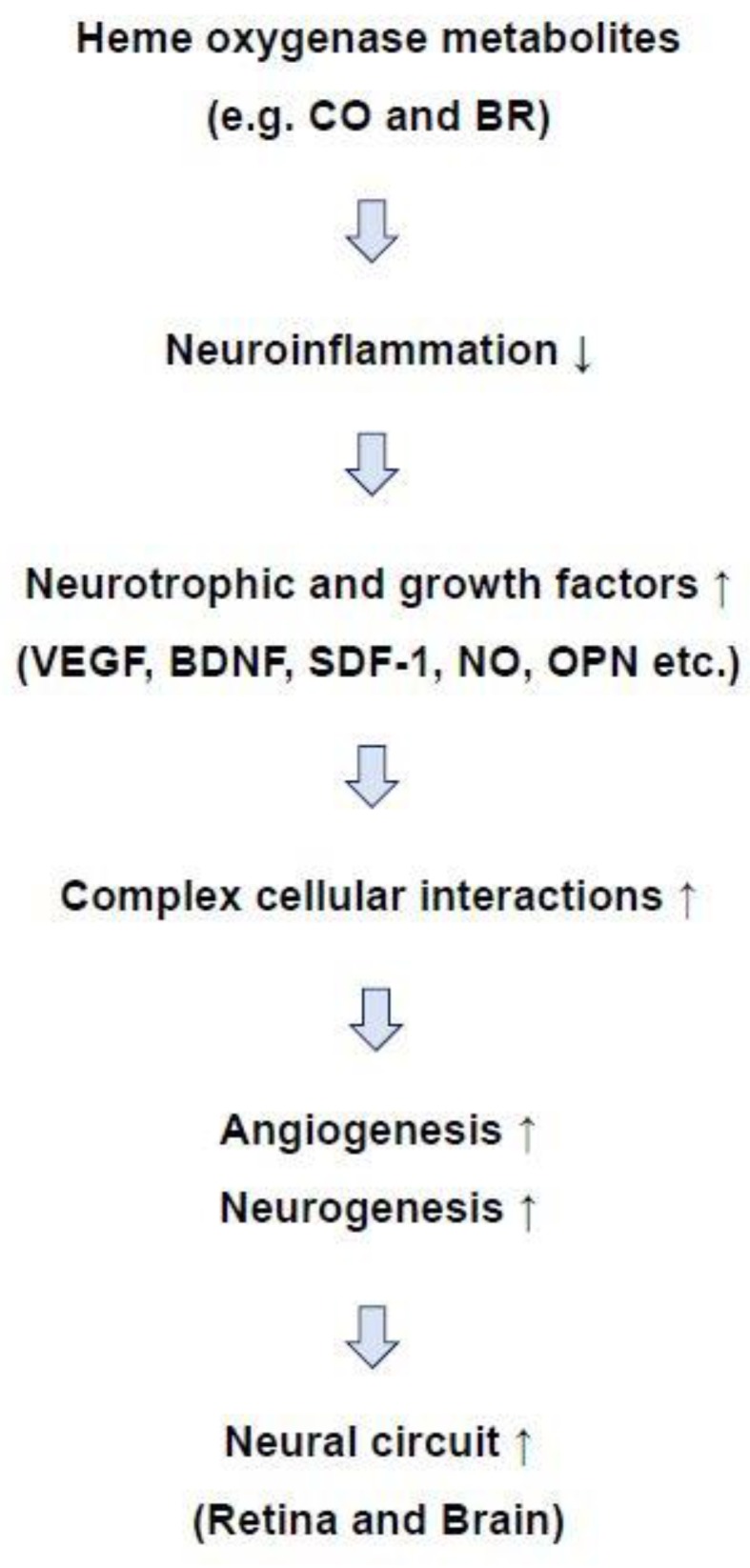
Possible role of heme oxygenase metabolites in a powerful self-repair mechanism. Carbon monoxide and bilirubin may reduce neuroinflammation and increase the secretion of neurotrophic and growth factors. Released factors can bind to their receptors in neighboring cells, such as endothelial cells and neural stem cells, enhancing cellular interactions. Generation of new vessels and differentiation of NSCs into mature neurons may repair neural circuits in the brain and retina.

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
