# Peer review of "Regenerative Effects of Heme Oxygenase Metabolites on Neuroinflammatory Diseases"

_ijms, 2018, doi:10.3390/ijms20010078_

Round 1
Reviewer 1 Report
In this manuscript, the Authors provided an overview about the role of heme oxygenase and its metabolites on neuroinflammatory diseases. The topic is hot and interesting, but several points need to be amended.
Heme oxygenase (HO) exists in two main isoforms, namely HO-1 and HO-2; HO-3 is considered a spliced-variant of HO-2 and does not have any enzymatic activity. HO-2 is constitutive (and not constant, see line 34) and serves as a sensor not only for oxygen, but also for other gaseous molecules. In addition, HO and the by-product carbon monoxide (CO), have been demonstrated, several years ago, to behave as pro-inflammatory molecules, since they blunt the adrenal response to stressors (Mancuso et al., J Neuroimmunol 1999; Mancuso et al., Neuroimmunomodulation 1997). In addition, and in support of the pro-inflammatory nature of HO, is the evidence that this latter stimulates prostaglandin synthesis in the rat brain (Mancuso, Pistritto et al., Mol Brain Res 1997; Mancuso et al., J Neurosci Res 2006). Several papers have also shown as HO-1 undergoes repression under strong and long-lasting pro-oxidant conditions in order to avoid toxic accumulation of both CO and ferrous iron (Palozza et al., Antioxid Redox Signal 2006; Shibahara et al., Exp. Biol Med 2003; Nakayama et al., BBRC 2000). A similar dual role, i.e. neuroprotective and neurotoxic, has been demonstrated for bilirubin (BR) (Mancuso, Neuropharmacology 2017). The Authors did not mention the link between BR and nitric oxide, quite important as a determinant of neuroinflammation (Mancuso et al., Neurosci Lett 2012; Mancuso et al., J Neurosci Res 2008; Mancuso et al., Antioxid Redox Signal 2006). Because of BR-NO interaction, the formation of N-nitro-BR has been also shown (Barone et al., J Cell Mol Med 2009). Since herpesviruses infection has been proposed as a potential mechanism underlying neuroinflammation, noteworthy is the study by Santangelo et al. (Front Pharmacol 2012) dealing with the antiviral activity of BR. The Authors did not mention at all the role played by both HO and biliverdin reductase (BVR) in Alzheimer’s disease (Barone et al., Free Rad Biol Med 2012; Barone et al., Biochim Biophys Acta 2011; Barone et al., J Alzheimers Dis 2011) as well as thedruggability of both HO and BVR as targets of atorvastatin (Barone et al., J Neurochem 2012; Barone et al., Int J Neuropsychopharmacol 2012). I strongly suggest the Authors to keep in mind all the suggestions provided above and modify the paper accordingly. |
Author Response
Response to Comments by Reviewer 1
Comment #1: Heme oxygenase (HO) exists in two main isoforms, namely HO-1 and HO-2; HO-3 is considered a spliced-variant of HO-2 and does not have any enzymatic activity. HO-2 is constitutive (and not constant, see line 34) and serves as a sensor not only for oxygen, but also for other gaseous molecules.
Response: Thank you for your comment. We have changed those sentences per your suggestion.
(Lines 34–37) [HO-2, a constitutive isoform of HO, is present in high levels in the liver, brain, and testes. In the brain, HO-2 functions as a sensor for oxygen as well as other gaseous molecules and regulates vascular function. HO-3 is a pseudogene found in the rat brain and does not have enzymatic activity.]
Comment #2: In addition, HO and the by-product carbon monoxide (CO), have been demonstrated, several years ago, to behave as pro-inflammatory molecules, since they blunt the adrenal response to stressors (Mancuso et al., J Neuroimmunol 1999; Mancuso et al., Neuroimmunomodulation 1997). In addition, and in support of the pro-inflammatory nature of HO, is the evidence that this latter stimulates prostaglandin synthesis in the rat brain (Mancuso, Pistritto et al., Mol Brain Res 1997; Mancuso et al., J Neurosci Res 2006). Several papers have also shown as HO-1 undergoes repression under strong and long-lasting pro-oxidant conditions in order to avoid toxic accumulation of both CO and ferrous iron (Palozza et al., Antioxid Redox Signal 2006; Shibahara et al., Exp. Biol Med 2003; Nakayama et al., BBRC 2000).
Response: We have added the effects of HO metabolites on the pro-inflammatory response.
(Lines 55–61) [However, HO and its metabolites are Janus-faced. In this regard, HO-1 in astrocytes of the aging and diseased central nervous system (CNS) can be an effector of deleterious stimuli, leading to neuronal injury [1]. Administration of the HO inducer hemin can increase pro-inflammatory prostaglandin E2 levels in rat hypothalamic explants and in primary cultures of rat hypothalamic astrocytes [2]. CO and BR may also have dual roles (i.e., pro-inflammatory and anti-inflammatory) in several different organs and tissues [3, 4], depending on the concentration and the signaling pathway involved.]
(Lines 182–189) [In reactive glia, astrocyte-specific overexpression of HO-1 in conditions of oxidative stress leads to neuronal injury by releasing neurotoxic molecules such as IL-1b and TNFa [1, 5]. In transgenic mice overexpressing human HO-1 in their astrocytes, HO metabolites promote mitochondrial sequestration of non-transferrin iron, oxidative stress-mediated substrate modification within the mitochondria, and subsequent mitophagy [5]. Co-culture of PC12 cells with HO-1 overexpressing astrocytes induced PC12 cell death, which was reduced by treatment with deferoxamine, an iron chelating agent [6]. Therefore, the neurotoxic role of HO-1 in oxidative stress-conditioned astrocytes may stem from excessive iron deposition.]
(Lines 427–430) [However, sustained upregulation of HO and its metabolites in the brain may exacerbate neural injury. Therefore, the disparate properties of HO and its metabolites under various neuropathological conditions must be carefully considered for clinical approaches.]
Comment #3: A similar dual role, i.e. neuroprotective and neurotoxic, has been demonstrated for bilirubin (BR) (Mancuso, Neuropharmacology 2017). The Authors did not mention the link between BR and nitric oxide, quite important as a determinant of neuroinflammation (Mancuso et al., Neurosci Lett 2012; Mancuso et al., J Neurosci Res 2008; Mancuso et al., Antioxid Redox Signal 2006). Because of BR-NO interaction, the formation of N-nitro-BR has been also shown (Barone et al., J Cell Mol Med 2009). Since herpesviruses infection has been proposed as a potential mechanism underlying neuroinflammation, noteworthy is the study by Santangelo et al. (Front Pharmacol 2012) dealing with the antiviral activity of BR.
Response: In accordance with your comment, we have added the link between BR and nitric oxide to the text.
(Line 217–223) [The link between BR and NO has been reported as a determinant of neuroinflammation [3]. BR upregulates the nNOS/NO axis in primary rat cerebellar granule neurons exposed to serum starvation or conditions of neurotrophin deficiency [3]. In addition, BR serves as an endogenous scavenger for RNS by denitrosylating the thiol group of proteins and non-protein molecules [7]. BR can be formed by the HO-BVR pathway, and HO is co-expressed with BVR in the brain [8]. Therefore, BR may reduce peroxynitrite production when the HO-BVR pathway is activated in the TBI brain. One possible target that may lead to activation of the HO-1-BVR pathway in TBI might be CO.]
Comment #4: The Authors did not mention at all the role played by both HO and biliverdin reductase (BVR) in Alzheimer’s disease (Barone et al., Free Rad Biol Med 2012; Barone et al., Biochim Biophys Acta 2011; Barone et al., J Alzheimers Dis 2011) as well as the druggability of both HO and BVR as targets of atorvastatin (Barone et al., J Neurochem 2012; Barone et al., Int J Neuropsychopharmacol 2012).
Response: We have added the role played by both HO and biliverdin reductase (BVR) in Alzheimer’s disease.
(Lines 285–304) [2.2.4. AD
In patients with AD, failure of recent memory and other intellectual functions is observed. Amyloid precursor protein (APP) generates the b-amyloid (Aβ) peptide, postulated to participate in the neurotoxicity found in AD. Neuronal loss and reactive astrocytes may be associated with Ab-peptide toxicity and the deposition of neurofibrillary tangles containing hyperphosphorylated tau in AD [1]. HO-2 interacts with APP, and APP inhibits HO activity [9]. Treatment of rat hippocampal neurons with CORM-2 protects them against Ab-induced toxicity [10]. A marked reduction in neuronal BR levels may be related to an increased sensitivity to H2O2-induced neurotoxicity in transgenic APPswe mice [9], suggesting that HO metabolites, such as CO and BR, may play cytoprotective roles in AD. Similar to TBI, pericyte loss is also promoted in AD, consequently leading to the accumulation of neurotoxic Ab in transgenic APPswe mice [11]. The link between HO metabolites and functional pericyte recovery is of great interest in AD.
BVRA can be modified by oxidative/nitrosative stress into tyrosine nitration (3-NT-BVRA), which is increased in the hippocampus of 12 and 18 month-old 3xTg-AD mice compared with wild-type mice [12]. The 3xTg-AD mice have three mutant human genes, i.e., APPswe, PS1M146V, and tauP301L, and show reductions in BVRA protein levels at 3, 6, and 12 months old [12]. Impaired BVRA activity, partly via 3-NT-BVRA, and reduction in BVRA expression may result in increased oxidative stress and inflammation in AD [12, 13]. Administration of the cholesterol lowering agent atorvastatin can induce BVRA and HO-1 protein levels [14], resulting in increases in CO and BR, consequently reducing oxidative stress-mediated neuroinflammation.]
References
1. Schipper, H. M.; Song, W.; Tavitian, A.; Cressatti, M., The sinister face of heme oxygenase-1 in brain aging and disease. Prog Neurobiol 2018.
2. Mancuso, C.; Perluigi, M.; Cini, C.; De Marco, C.; Giuffrida Stella, A. M.; Calabrese, V., Heme oxygenase and cyclooxygenase in the central nervous system: a functional interplay. J Neurosci Res 2006, 84, (7), 1385-91.
3. Mancuso, C., Bilirubin and brain: A pharmacological approach. Neuropharmacology 2017, 118, 113-123.
4. Barone, E.; Di Domenico, F.; Mancuso, C.; Butterfield, D. A., The Janus face of the heme oxygenase/biliverdin reductase system in Alzheimer disease: it's time for reconciliation. Neurobiol Dis 2014, 62, 144-59.
5. Song, W.; Cressatti, M.; Zukor, H.; Liberman, A.; Galindez, C.; Schipper, H. M., Parkinsonian features in aging GFAP.HMOX1 transgenic mice overexpressing human HO-1 in the astroglial compartment. Neurobiol Aging 2017, 58, 163-179.
6. Li, M. H.; Jang, J. H.; Na, H. K.; Cha, Y. N.; Surh, Y. J., Carbon monoxide produced by heme oxygenase-1 in response to nitrosative stress induces expression of glutamate-cysteine ligase in PC12 cells via activation of phosphatidylinositol 3-kinase and Nrf2 signaling. J Biol Chem 2007, 282, (39), 28577-86.
7. Barone, E.; Trombino, S.; Cassano, R.; Sgambato, A.; De Paola, B.; Di Stasio, E.; Picci, N.; Preziosi, P.; Mancuso, C., Characterization of the S-denitrosylating activity of bilirubin. J Cell Mol Med 2009, 13, (8B), 2365-75.
8. Ewing, J. F.; Weber, C. M.; Maines, M. D., Biliverdin reductase is heat resistant and coexpressed with constitutive and heat shock forms of heme oxygenase in brain. Journal of neurochemistry 1993, 61, (3), 1015-23.
9. Takahashi, M.; Dore, S.; Ferris, C. D.; Tomita, T.; Sawa, A.; Wolosker, H.; Borchelt, D. R.; Iwatsubo, T.; Kim, S. H.; Thinakaran, G.; Sisodia, S. S.; Snyder, S. H., Amyloid precursor proteins inhibit heme oxygenase activity and augment neurotoxicity in Alzheimer's disease. Neuron 2000, 28, (2), 461-73.
10. Hettiarachchi, N.; Dallas, M.; Al-Owais, M.; Griffiths, H.; Hooper, N.; Scragg, J.; Boyle, J.; Peers, C., Heme oxygenase-1 protects against Alzheimer's amyloid-beta(1-42)-induced toxicity via carbon monoxide production. Cell Death Dis 2014, 5, e1569.
11. Sagare, A. P.; Bell, R. D.; Zhao, Z.; Ma, Q.; Winkler, E. A.; Ramanathan, A.; Zlokovic, B. V., Pericyte loss influences Alzheimer-like neurodegeneration in mice. Nat Commun 2013, 4, 2932.
12. Barone, E.; Di Domenico, F.; Cassano, T.; Arena, A.; Tramutola, A.; Lavecchia, M. A.; Coccia, R.; Butterfield, D. A.; Perluigi, M., Impairment of biliverdin reductase-A promotes brain insulin resistance in Alzheimer disease: A new paradigm. Free radical biology & medicine 2016, 91, 127-42.
13. Chen, W.; Maghzal, G. J.; Ayer, A.; Suarna, C.; Dunn, L. L.; Stocker, R., Absence of the biliverdin reductase-a gene is associated with increased endogenous oxidative stress. Free radical biology & medicine 2018, 115, 156-165.
14. Barone, E.; Di Domenico, F.; Butterfield, D. A., Statins more than cholesterol lowering agents in Alzheimer disease: their pleiotropic functions as potential therapeutic targets. Biochemical pharmacology 2014, 88, (4), 605-16.

Reviewer 2 Report
1. This topic is important in so far as the heme oxygenases have been increasingly implicated in the redox neurobiology of numerous neuroinflammatory and neurodegenerative disorders.
2. The phraseology occasionally seems awkward, reflecting the fact that English is likely not the authors’ primary language. For example, on p. 1, I recommend changing “constant form of HO isoforms” to “constitutive isoform of heme oxygenase”; on p.6, “the mouse blinded model” should be re-phrased.
3. On p. 1, the authors state that “HO-3, another form of HO, is found in the rat brain”. More accurately, HO-3 is a pseudogene (retrotransposition of Hmox2) specific to rats (Scapagnini et al., 2002).
4. This reviewer’s most pressing concern is that the authors entirely neglected to present the other ‘Janus’ face of HO-1—viz., that sustained up-regulation of HO-1 in brain may exacerbate, rather than mitigate, neural injury in a host of human neurological conditions. A balanced literature should acknowledge that, in a host of chronic human CNS afflictions, the glial HO-1 response may serve as a transducer of noxious stimuli and an important driver of relevant neuropathology. Specifically, sustained over-expression of HO-1 in astroglia, with attendant liberation of intracellular free iron and CO, may contribute to the pathological iron deposition, oxidative damage and mitochondrial insufficiency documented in multiple sclerosis – an important neuroimmunological disorder - as well as several neurodegenerative conditions such as Alzheimer disease and Parkinson disease (recently reviewed in Prog Neurobiol, in press (2018); DOI: 10.1016/j.pneurobio.2018.06.008). On p. 8 the authors state that “HO-1 overexpression increased the survival rate of dopaminergic neurons” in an MPP+ rat model of Parkinson’s disease”. Yet, the authors fail to mention that overexpression of astroglial HO-1 (to levels seen in human PD brain) in middle-aged mice results in a robust phenotype highly reminiscent of idiopathic PD (Neurobiol Aging 58: 163-179, 2017). The disparate behaviour of HO-1 under various neuropathological conditions may be reconciled by the fact that intracellular heme degradation may exert net antioxidant or pro-oxidant effects contingent upon the intensity and temporal profile of HO-1 induction and its interplay with the prevailing redox microenvironment. The authors should seriously consider the ongoing controversy regarding the potential benefits and liabilities of chronic HO-1 expression in brain before advocating induction of the enzyme as a therapeutic modality in neuroinflammatory and other CNS conditions.
Author Response
Response to Comments by Reviewer 2
Comment #1: The phraseology occasionally seems awkward, reflecting the fact that English is likely not the authors’ primary language. For example, on p. 1, I recommend changing “constant form of HO isoforms” to “constitutive isoform of heme oxygenase”; on p.6, “the mouse blinded model” should be re-phrased.
Response: We have changed those sentences per your suggestion.
(Lines 34–35) [HO-2, a constitutive isoform of HO, is present in high levels in the liver, brain, and testes.]
(Lines 254–255) [Müller cells can be reprogrammed to generate rod photoreceptors, leading to restored visual responses in a mouse model of congenital blindness]
Comment #2: On p. 1, the authors state that “HO-3, another form of HO, is found in the rat brain”. More accurately, HO-3 is a pseudogene (retrotransposition of Hmox2) specific to rats (Scapagnini et al., 2002).
Response: We have changed that sentence per your suggestion.
(Lines 36–37) [HO-3 is a pseudogene found in the rat brain and does not have enzymatic activity [1, 2].]
Comment #3: This reviewer’s most pressing concern is that the authors entirely neglected to present the other ‘Janus’ face of HO-1—viz., that sustained up-regulation of HO-1 in brain may exacerbate, rather than mitigate, neural injury in a host of human neurological conditions. A balanced literature should acknowledge that, in a host of chronic human CNS afflictions, the glial HO-1 response may serve as a transducer of noxious stimuli and an important driver of relevant neuropathology. Specifically, sustained over-expression of HO-1 in astroglia, with attendant liberation of intracellular free iron and CO, may contribute to the pathological iron deposition, oxidative damage and mitochondrial insufficiency documented in multiple sclerosis – an important neuroimmunological disorder - as well as several neurodegenerative conditions such as Alzheimer disease and Parkinson disease (recently reviewed in Prog Neurobiol, in press (2018); DOI: 10.1016/j.pneurobio.2018.06.008). On p. 8 the authors state that “HO-1 overexpression increased the survival rate of dopaminergic neurons” in an MPP+ rat model of Parkinson’s disease”. Yet, the authors fail to mention that overexpression of astroglial HO-1 (to levels seen in human PD brain) in middle-aged mice results in a robust phenotype highly reminiscent of idiopathic PD (Neurobiol Aging 58: 163-179, 2017). The disparate behaviour of HO-1 under various neuropathological conditions may be reconciled by the fact that intracellular heme degradation may exert net antioxidant or pro-oxidant effects contingent upon the intensity and temporal profile of HO-1 induction and its interplay with the prevailing redox microenvironment. The authors should seriously consider the ongoing controversy regarding the potential benefits and liabilities of chronic HO-1 expression in brain before advocating induction of the enzyme as a therapeutic modality in neuroinflammatory and other CNS conditions.
Response: We changed these sentences per your suggestion.
(Lines 55–61) [However, HO and its metabolites are Janus-faced. In this regard, HO-1 in astrocytes of the aging and diseased central nervous system (CNS) can be an effector of deleterious stimuli, leading to neuronal injury [3]. Administration of the HO inducer hemin can increase pro-inflammatory prostaglandin E2 levels in rat hypothalamic explants and in primary cultures of rat hypothalamic astrocytes [4]. CO and BR may also have dual roles (i.e., pro-inflammatory and anti-inflammatory) in several different organs and tissues [5, 6], depending on the concentration and the signaling pathway involved.]
(Lines 182–189) [In reactive glia, astrocyte-specific overexpression of HO-1 in conditions of oxidative stress leads to neuronal injury by releasing neurotoxic molecules such as IL-1b and TNF-a [3, 7]. In transgenic mice overexpressing human HO-1 in their astrocytes, HO metabolites promote mitochondrial sequestration of non-transferrin iron, oxidative stress-mediated substrate modification within the mitochondria, and subsequent mitophagy [7]. Co-culture of PC12 cells with HO-1 overexpressing astrocytes induced PC12 cell death, which was reduced by treatment with deferoxamine, an iron chelating agent [8]. Therefore, the neurotoxic role of HO-1 in oxidative stress-conditioned astrocytes may stem from excessive iron deposition.]
(Lines 427–430) [However, sustained upregulation of HO and its metabolites in the brain may exacerbate neural injury. Therefore, the disparate properties of HO and its metabolites under various neuropathological conditions must be carefully considered for clinical approaches.]
References
1. Rochette, L.; Zeller, M.; Cottin, Y.; Vergely, C., Redox Functions of Heme Oxygenase-1 and Biliverdin Reductase in Diabetes. Trends Endocrinol Metab 2018, 29, (2), 74-85.
2. Scapagnini, G.; D'Agata, V.; Calabrese, V.; Pascale, A.; Colombrita, C.; Alkon, D.; Cavallaro, S., Gene expression profiles of heme oxygenase isoforms in the rat brain. Brain Res 2002, 954, (1), 51-9.
3. Schipper, H. M.; Song, W.; Tavitian, A.; Cressatti, M., The sinister face of heme oxygenase-1 in brain aging and disease. Prog Neurobiol 2018.
4. Mancuso, C.; Perluigi, M.; Cini, C.; De Marco, C.; Giuffrida Stella, A. M.; Calabrese, V., Heme oxygenase and cyclooxygenase in the central nervous system: a functional interplay. J Neurosci Res 2006, 84, (7), 1385-91.
5. Mancuso, C., Bilirubin and brain: A pharmacological approach. Neuropharmacology 2017, 118, 113-123.
6. Barone, E.; Di Domenico, F.; Mancuso, C.; Butterfield, D. A., The Janus face of the heme oxygenase/biliverdin reductase system in Alzheimer disease: it's time for reconciliation. Neurobiol Dis 2014, 62, 144-59.
7. Song, W.; Cressatti, M.; Zukor, H.; Liberman, A.; Galindez, C.; Schipper, H. M., Parkinsonian features in aging GFAP.HMOX1 transgenic mice overexpressing human HO-1 in the astroglial compartment. Neurobiol Aging 2017, 58, 163-179.
8. Li, M. H.; Jang, J. H.; Na, H. K.; Cha, Y. N.; Surh, Y. J., Carbon monoxide produced by heme oxygenase-1 in response to nitrosative stress induces expression of glutamate-cysteine ligase in PC12 cells via activation of phosphatidylinositol 3-kinase and Nrf2 signaling. J Biol Chem 2007, 282, (39), 28577-86.

Round 2
Reviewer 1 Report
In this revised manuscript, the Authors included only some recommendations provided in the first round of revision.
Although my previous recommendation, the Authors did not detailed the pro-inflammatory role of CO; in lines 59-61 only a brief mention has been done and this is not balanced because, throughout the paper, the Authors continued to magnify the anti-inflammatory effects of CO. The Authors must provide details about the ability of CO to blunt the hypothalamus-pituitary-adrenal axis (see previous comments). Furthermore, several mistakes have been spotted among the references. Reference #11 and #12 do not refer directly to CO's pro-inflammatory activity; the Authors may want to take advantage of reading previous comments and cite proper references.
The interaction between BR and NO has not been adequately addressed (see previous comments). I do not understand the reason why the Authors decided to include preclinical data on HO-1 and BVR post-translational modifications in AD the presence of specific data obtained in human samples (brain and serum/plasma): the Authors should take into proper consideration this suggestion. Reference #76 refers to a wide review on the neuroprotective effects of statins; once again, my suggestion is to provide references of original articles dealing with this issue (see previous recommendations). Finally, as recommended also by Reviewer #2, the repression of HO-1 gene must be deeply addressed (see previous comments).
Author Response
Response to Reviewer 1
Comment #1: Although my previous recommendation, the Authors did not detailed the pro-inflammatory role of CO; in lines 59-61 only a brief mention has been done and this is not balanced because, throughout the paper, the Authors continued to magnify the anti-inflammatory effects of CO.
Response: We have added those sentences per your suggestion.
(Lines 69–78) [CO and BR may also play dual roles (i.e., pro-inflammatory and anti-inflammatory) in a variety of organs and tissues [13, 14], depending on their concentration and the signaling pathways involved. Sustained activation of the HO-1-CO pathway may facilitate the development of neuroendocrine disturbances characteristic of age-related neuroinflammatory diseases [15]. To prevent neurotoxicity, BR must be glucuronidated and excreted in the bile. An excessive accumulation of BR in the brain would result in kernicteric damage. Similar to CO and BR, free iron has a pro-oxidant capacity in a redox-active form, leading to lipid, protein, and DNA damage. Overexpression of the HO-1 gene in oxidatively stressed astroglial cells may perpetuate intracellular reactive oxygen species (ROS) generation, oxidative mitochondrial injury, and non-transferrin-derived iron deposition within the mitochondrial compartment [16].]
Comment #2: The Authors must provide details about the ability of CO to blunt the hypothalamus-pituitary-adrenal axis (see previous comments).
Response: We have added details regarding the ability of CO to blunt the hypothalamus-pituitary-adrenal axis.
(Lines 59–68) [On the other side, the HO inhibitor Sn-protoporphyrin-9 has been proven to amplify the significant activation of the hypothalamus-pituitary-adrenal axis induced by bacterial lipopolysaccharide administration in male Wistar rats [11]. This observation indicates a protective role for HO in counteracting potentially dangerous surges of serum vasopressin levels, leading to hypothalamic vasopressin depletion. Accordingly, in in vitro studies, the formation of CO within the hypothalamus has been associated with inhibition of the release of hormones, such as corticotropin-releasing hormone, arginine vasopressin and oxytocin, involved in hypothalamus-pituitary-adrenal axis activation [12]. These findings suggest that the HO-CO pathway may have a neuroendocrine modulatory role, preventing over-exuberant activation of the hypothalamus-pituitary-adrenal axis during stress.]
Comment #3: Furthermore, several mistakes have been spotted among the references. Reference #11 and #12 do not refer directly to CO's pro-inflammatory activity; the Authors may want to take advantage of reading previous comments and cite proper references.
Response: In accordance to the reviewer’s advice, we have changed the needed References.
(Line 69–70) [CO and BR may also play dual roles (i.e., pro-inflammatory and anti-inflammatory) in a variety of organs and tissues [13, 14], depending on their concentration and the signaling pathways involved.]
Comment #4: The interaction between BR and NO has not been adequately addressed (see previous comments).
Response: We have added the interaction between BR and NO per your suggestion.
(Lines 234–249) [In the absence of exogenous stimuli, BR upregulates the phosphorylation of the cAMP responsive element binding (CREB) factor and the production of NO. The extracellular Ca2+ chelator ethylene glycol-bis(2-aminoethylether)-N,N,N′,N′-tetraacetic acid (EGTA) interferes with this pathway [62], suggesting a role for BR in Ca2+-mediated CREB and nNOS activation. Both CREB and NO are considered important factors contributing to synaptic plasticity and memory consolidation by regulating the expression of brain-derived neurotrophic factor (BDNF) [63, 64]. Therefore, BR may boost the repair process counteracting the deficiency of neurotrophic factors following brain injury. On the other hand, treatment of PC12 cells with BDNF or neurotrophic growth factor increased the signaling to Akt (Protein Kinase B) and extracellular signal-regulated kinases (ERKs), which are crucial factors for survival, and these effects were markedly reduced by BR [62]. Therefore, these observations indicate an important action of BR on survival signaling mediated by neurotrophins, with either inhibitory or agonistic effects based on growth factor availability.
In addition, BR serves as an endogenous scavenger for RNS by denitrosylating the thiol group of proteins and non-protein molecules [65]. In response to exogenous hydrogen peroxide, BR markedly decreases ROS generation in PC12 cells [62].]
Comment #5: I do not understand the reason why the Authors decided to include preclinical data on HO-1 and BVR post-translational modifications in AD the presence of specific data obtained in human samples (brain and serum/plasma): the Authors should take into proper consideration this suggestion.
Response: We have added the data showing the expression of both HO and biliverdin reductase (BVR) in Alzheimer’s disease.
(Lines 325–338) [Recently, a down-regulation of HO-2 and BVR has been demonstrated in post-mortem brain tissues of AD subjects as compared to tissues of age-matched controls. These changes were found in the hippocampus, an area associated with cognitive functions such as learning and memory [82-84]. Moreover, a significant increase in the phosphorylation of HO-1 serine residues was observed in the hippocampus of AD subjects [84]. As a result of phosphorylation, HO-1 can become a target for oxidative posttranslational modifications of the protein structure, with consequent functional impairment. In addition, repression of HO-1 mRNA has been reported under strong and sustained pro-oxidant conditions such as hypoxia, heat shock, or interferon-γ [85-87]. Treatment of HUVECs with the iron chelator, desferrioxamine, or hypoxia reduced HO-1 mRNA expression [87], possibly preventing toxic accumulation of HO metabolites such as iron and CO. Though HO-1 mRNA expression in plasma has been reported to be markedly reduced in AD subjects as compared to healthy controls [88], recent studies show enhanced HO-1 protein levels in plasma, hippocampus, and cerebellum of AD subjects [84, 89]. Therefore, the reports of HO-1 levels in AD are still controversial.]
Comment #6: Reference #76 refers to a wide review on the neuroprotective effects of statins; once again, my suggestion is to provide references of original articles dealing with this issue (see previous recommendations).
Response: We have added references to the original experiments.
(Lines 345–351) [Administration of the cholesterol-lowering agent atorvastatin can induce BVRA and HO-1 protein expression in the parietal cortex of aging dogs [91-93], resulting in increased BR levels. In addition, atorvastatin lowered oxidative/nitrosative stress biomarkers, such as 3-nitrotyrosine, in the same cortical area [92, 93], leading to the reduction of oxidative stress-mediated neuroinflammation. Size discrimination learning error scores negatively correlated with BVRA protein levels and BVR activity [92]. Therefore, atorvastatin-mediated HO-1 and BVR upregulation may be associated with reduced oxidative stress and improved cognitive functions in the aging brain.]
Comment #7: Finally, as recommended also by Reviewer #2, the repression of HO-1 gene must be deeply addressed (see previous comments).
Response: We have added the repression of HO-1 gene under oxidative stress conditions.(Lines 331–334) [In addition, repression of HO-1 mRNA has been reported under strong and sustained pro-oxidant conditions such as hypoxia, heat shock, or interferon-γ [15-17]. Treatment of HUVECs with the iron chelator, desferrioxamine, or hypoxia reduced HO-1 mRNA expression [17], possibly avoiding toxic accumulation of HO metabolites such as iron and CO.]

Reviewer 2 Report
The authors have satisfactorily responded to the comments/suggestions of this reviewer. One further emendation is required: On Line 183, the authors cite sources indicating that IL-1β and TNFα are released from HO-1 overexpressing astrocytes. Actually, those studies showed that IL-1β and TNFα induce Hmox1 in astrocytes which, in turn, damages mitochondria.
Author Response
Response to Comments by Reviewer 2
Comment #1: Finally, One further emendation is required: On Line 183, the authors cite sources indicating that IL-1β and TNFα are released from HO-1 overexpressing astrocytes. Actually, those studies showed that IL-1β and TNFα induce Hmox1 in astrocytes which, in turn, damages mitochondria.
Response: We have changed those sentences per your suggestion.
(Lines 197–201) [In reactive glia, the increase in neurotoxic molecules, such as IL-1b and TNF-a, induces HO-1 mRNA expression in conditions of oxidative stress which, in turn, leads to mitochondrial damage [9, 50]. In transgenic mice overexpressing human HO-1 in their astrocytes, HO metabolites promote mitochondrial sequestration of non-transferrin iron, oxidative stress-mediated substrate modification within the mitochondria, and subsequent mitophagy [50].]

Round 3
Reviewer 1 Report
This reviewer wants to congratulate the Authors for the nice work done in this second round of revision. The quality of the manuscript is now greatly increased.